# How to Protect the Credibility of Articles Published in Predatory Journals

Yuki Yamada 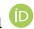

Faculty of Arts and Science, Kyushu University, Fukuoka 819-0395, Japan; yamadayuk@gmail.com;
Tel.: +81-92-802-5837

**Abstract:** Predatory journals often prey on innocent researchers who are unaware of the threat they pose. This paper discusses what researchers can do if they unintentionally publish a paper in a predatory journal, including measures to take before submission, during peer review, and after the journal has accepted a manuscript. The specific recommendations discussed are pre-registration, pre-submission peer-review, open peer-review, topping up reviewers, post-publication peer review, open recommendation, and treatment as unrefereed. These measures may help to ensure the credibility of the article, even if it is published in a predatory journal. The present article suggests that an open and multi-layered assessment of research content enhances the credibility of all research articles, even those published in non-predatory journals. If applied consistently by researchers in various fields, the suggested measures may enhance reproducibility and promote the advancement of science.

**Keywords:** publication bias; predatory journals/publishing; research personnel; quality control; periodicals; peer review; research; inexperienced researchers; ethical publishing

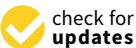



## 1. Introduction

For researchers, publishing a peer-reviewed research paper in an academic journal is critical to their professional life—often referred to in academia as "publish-or-perish". However, submitting a manuscript to a journal does not necessarily lead to publication. A study author may equate failure to publish to their "death" as a researcher. The corresponding pressure on academic professionals to be published create factors such as publication bias (i.e., a bias in which only manuscripts with favorable and generally positive results for the author are published) [1,2], and questionable research practices (QRPs) [3,4] that distort the reliability of scientific endeavor [5].

Reputable journals are likely to be highly selective in the submissions they select for publication: At times, manuscripts are rejected simply because they are outside the scope of the journals. Moreover, the bandwidth of the peer-review "filter" varies from journal to journal. If a particular journal's peer-review filter is less rigorous, then the natural progression for researchers whose careers may suffer if their work is not published is to improve their chances by submitting their articles to such a journal. At the extreme are journals that have low or no vetting of content. It is under this set of circumstances that business opportunities arise. Such so-called predatory journals falsely claim to provide peer-review or provide only very superficial peer review (although there are some predatory journals that provide peer review of unknown quality [6]), accepting many papers that might not pass peer review by reputable journals. In fact, about one-third of the authors of predatory journals had been previously rejected and, of that group, many (43%) had been rejected twice [6]. In this way, predatory journals (and publishers, henceforth omitted) can profit from collecting publication fees from many researchers seeking to avoid "dying" within their field. Peer review is minimal or absent in these journals, but the manuscript is always accepted for publication. Acceptance rates have been estimated at 80–100%, depending on the specific journal [7]). Since unrefereed papers are published as refereed

papers, publication in such journals is often called "resumé padding," [8] and the studies are sometimes deemed "junk science" [9]. Furthermore, because publishing fees paid to predatory journals may be paid by public research funds, predatory publishing has been criticized [8] and is regarded as unethical [10] for wasting taxpayer money and other economic resources of stakeholders.

In some cases, a researcher may work hard to prepare a high-quality manuscript, but if they are inexperienced, they might not be well-informed about predatory publishing, especially in universities/faculties with insufficient and inadequate supervision and mentoring [11]. These authors may be unable to choose legitimate journals, and as a result, they unintentionally submit the paper they have prepared and heavily invested in to a predatory journal (novice and other authors can benefit from https://thinkchecksubmit.org/, which helps empower authors to make informed choices for publication and avoid predatory publishers). The authors may subsequently realize that the journal to which they submitted their work is predatory, but they could be left at a loss, as predatory journals often do not readily accept an author's request for withdrawal/retraction of articles [12]. If they do, they never refund the publication fees [13]. Rather, it is common to ask for a "withdrawal fee" of several hundred dollars, something that Beall described as holding the manuscript "hostage" until the ransom is paid [14]. This difficult issue of withdrawal and refund has been discussed by Culley and others [15].

Moreover, authors may worry that their reputation will suffer if other researchers were to find out that they had published a paper in a predatory journal. In fact, some institutions negatively assess researchers who have published in certain journals [16]. My institution (Kyushu University, Fukuoka, Japan) explicitly discourages faculty and staff from submitting to predatory journals. One researcher reportedly removed the history of those papers published in predatory journals from their CV. At least for me[1], this action may be considered a QRP, as it is a type of research achievement falsification[2]. Hiding achievements that would be detrimental to an individual can unfairly enhance their reputation. In general, if a job seeker does not write about the loss of social and academic reputation on a résumé, it is résumé fraud. Rather than simply hiding a disadvantageous paper, efforts should be made to withdraw or improve its credibility, as it is already published.

On the other hand, there are authors who intentionally publish in predatory journals, willingly using such journals for their self-interest (academic promotion, academic incentive, fear of job loss, or for grant applications) [17]. The worrying problem in academia is how easily such complicity can be achieved. As a result of our existing quantity-driven bibliometric culture, assessors may gloss over publications in predatory journals. Psychological discussions about the motivational aspects of such authors and institutional discussions, including penalties, are outside the scope of this paper, but they are important topics. Such intentionally predatory authors will continue to emerge, as long as the publish-or-perish situation persists.

How then, if we have unintentionally published in predatory journals, can we ensure the credibility of a paper without denying the fact that it has been published in a predatory journal? Authors in this situation may have recourse. Arguably, the biggest problem in publishing in predatory journals is that their internal assessment systems are dysfunctional or non-existent; therefore, to offset this problem, authors must compensate in some way.

---

[1]　One reviewer commented on this topic as follows. "If many authors publish in a predatory journal by accident, then there is no intentional research misconduct. I would be far less forceful about this statement. Some institutions require authors to remove predatory publications from their CV (e.g., my institution). I think the answer to this problem is unclear and may be an individual judgment call. It is not black and white." I can understand this sentiment. Although some researchers may be required to obey an institution's order to erase the history of their papers in predatory journals, I believe that this situation should eventually be resolved by making everything public. However, as this reviewer says, the answer to this problem remains unclear, so I have provided both sides of this point here.

[2]　Of course, omission is not always unethical. Sometimes omissions are used for clarity or brevity (i.e., omissions of unimportant facts). However, evidence of publication in predatory journals is important for confirming the ethics of the researcher, and this information should not be omitted. Ideally, there should be a database that ensures that all researchers' publications are included and made public. Currently, ORCID may not include predatory journals. For this reason, the completeness and openness of bibliographic information and its active use for researcher evaluation should be better encouraged in the future.

Herein are some ways for researchers to address this important issue at each stage in the publishing process—before submitting a paper, during peer review, and after publication. However, as this article concerns papers published in predatory journals, it does not include ways to avoid such journals or withdraw/retract submitted papers. Furthermore, the methods I present here are not directed at authors who intentionally published in predatory journals at all, since there is no need to save such authors.

## 2. Prior to Submission

Researchers typically do not intend to submit to a predatory journal before their paper has undergone peer review. Nevertheless, there are several things that all researchers should do prior to submission.

### 2.1. Pre-Registration

Registering hypotheses, experimental and analytical methods, etc., with a third-party organization, such as the Open Science Framework before conducting research can reduce the degree of freedom for researchers and increase the credibility of their research results. The registration is not required to be peer-reviewed by a third party and is completed when the information is registered by the researcher. Once registered, the information is confirmed with a time stamp and cannot be modified or changed. This system, called pre-registration [18,19], has been used in the medical field for a long time, and it has become more popular in other fields (e.g., psychology) in recent years. Pre-registration is often effective, even if a researcher's work has been published in predatory journals. While predatory journals are, in a sense, helping to deter publication bias in that they publish study articles regardless of the results, the problem lies in the credibility of the results. Pre-registration adds a certain degree of credibility to the results and their interpretation by predetermining and disclosing the research protocol. Pre-registration is part of the TOP (Transparency and Openness Promotion) guidelines proposed by the Center for Open Science. In addition to pre-registration, it includes citation standards; transparency of data, code, materials, design, and analytical plan; and replication, all of which contribute significantly to the credibility and reproducibility of scientific research. Researchers should pre-register regardless of whether the study is going to be submitted to predatory journals. However, it should be noted that pre-registration itself can be manipulated [20,21].

### 2.2. Paid Pre-Submission Peer Review

In principle, peer review can be performed at any time and can, of course, be done before a manuscript is submitted [22,23]. Some commercial companies, mostly those that provide English language manuscript editing, offer pre-submission peer review services for a fee. Authors can obtain at least minimal proof of peer review by having their manuscripts reviewed by such a service and then attaching the log provided in a supplementary section or including a link to the repository in the manuscript body text. However, as predatory journals often do not have a supplementary section, the use of a repository is recommended. For example, this manuscript has been professionally pre-reviewed for a fee, and a link to the review log is provided here (https://osf.io/kps75/).

The advantage of paid, commercial peer review is that it is decoupled from journals. In other words, it suppresses the bias in peer review comments that reviewers selected from journals will knowingly or unknowingly follow the journal's policies and reputation. Furthermore, since the review comments are given to the article, not to the journal, in principle, it is available to any editor. This is also the idea behind the community peer-review process described below, which is used as a streamlined peer-review system. On the other hand, its disadvantage is that the quality of peer review is unknown. In addition, disparities among researchers may occur due to economic reasons, as researchers must have sufficient funds to adopt this practice. This leads to the Matthew effect, where the rich get richer [24,25].

### 2.3. Pre-Submission Community Peer-Review

Some researcher communities such as "Review Commons (https://www.reviewcommons.org/) and "Peer Community In" (PCI: https://peercommunityin.org/), conduct pre-submission peer review of manuscripts, especially on preprint servers. Preprints are the authors' copy of the manuscript that has not yet been formally published in a journal. Many journals consider them unpublished and do not include them in double submission. However, there is an objection that preprints are considered to be published, rather than unpublished, papers [26] and some journals do not accept submissions of preprints. These communities provide a quality certification for preprints that pass their peer review. In particular, PCI recommends acceptance without further peer review for affiliated journals.

While bona fide researchers are unlikely to prepare their manuscript with the intention of publishing it in a predatory journal, the information introduced here is relevant to all research. Thus, it is recommended as good practice that research authors review this information regularly, regardless of whether a paper is being submitted to a predatory journal.

## 3. During the Review Process

It is during the peer review stage that most authors realize that they have submitted their work to a predatory journal. Signs that a journal is predatory can be seen in several aspects of the publication process, but the peer review process alone is sufficient to enhance that conviction. If two or more of the following are true, the journal is extremely likely to be predatory: Peer review is completed unusually quickly[3]; the reviewers are clearly non-specialists (because they are often staff members of the predatory publisher); there are unusually few review comments or feedback is largely copyediting, if any (i.e., desk acceptance); or the editor's judgment is unnaturally lenient. Undertaking the following measures will allow research authors to operate with an attitude of good faith toward peer review, which will lead to higher credibility.

### 3.1. Open Peer Review

In recent years, an increasing number of journals have begun to disclose the peer review communications after the paper is published. "Open peer review" includes seven elements of openness in its definition (such as the identity and participation of reviewers), and publication of peer review reports is one of them [27]. Open peer review can occur before, during, and after the peer review process and includes the post-publication peer review described below. In a survey of 1500 reputable journals in 2019, open peer review (strictly speaking, open identities) was introduced in only 1% of those journals [28], making it far from being a major method yet. A major reason for this stagnation, as has been pointed out before [29], is that many experts are reluctant to become peer reviewers because of the overestimation of the potential risks of revealing their identities in an open peer review system.

Here, I focus on the open reports, that is, disclosure of peer review content. The hallmark of predatory journals is, after all, disguising the peer reviews. Therefore, keeping the peer-review content open allows readers to see and know with certainty whether peer-review has indeed been carried out on the paper. This provision also sends an important signal to third parties about the journal's level of predation. In contrast, any peer review that is not open may be potentially subject to low rigor, even for legitimate journals (e.g., [30]). I have experienced open peer review in all capacities as an author, reviewer, and editor. In closed peer review, there were certain instances where reviewers used offensive language or made questionable suggestions to increase the sample size after reporting experimental results (i.e., $p$-hacking: [4]), even in legitimate journals. However, I have not experienced this in open peer review thus far. Likewise, we expect that very low quality, practically non-existent peer review will also cease to occur in the open peer review system. Note that this paper has undergone an open peer review.

---

3  There are quality publishers that also provide rapid peer review (e.g., PLoS ONE).

*3.2. Additional Reviewer Request*

Predatory journals are not averse to peer review, but they are afraid of losing their source of income in submitted papers, by issuing rejections. If they know that their authors intend to publish the paper regardless of the peer review results, they may add reviewers on demand. Naturally, the more peer reviewers there are, the more likely quality control of the paper will be achieved (but predatory journals are also prone to fabricating peer reviews). These measures are the exact opposite of the traditional expectations of authors who want a confidential peer-review process and feedback from as few peer reviewers as possible, but the effect of gaining credibility is significant. Before submitting, I considered requesting additional peer reviewers for the present opinion piece if there was less than one peer reviewer, but as it is clear from the published peer review history, there was no need to do that because this manuscript had two expert reviewers who did an enthusiastic review.

## 4. After Acceptance

*4.1. Post-Publication Peer Review*

Peer review is not limited to pre-publication. While only a few reviewers typically assess a paper before it is published, after it is published, any reader may evaluate it. Some journals, including those published by F1000Research and PLoS, have (or had) a comments section (i.e., "open evaluation"), and such post-publication comments should be treated in the same way as citable peer reviews [31,32]. For example, some PLoS journals are beginning to treat public comments on manuscripts submitted simultaneously to bioRxiv as formal peer-review comments [33]. Thus, in theory, the quality of articles published in predatory journals may also be guaranteed by inviting comments after publication. If the journal does not have a comments section, it is advisable to release the accepted manuscript on a preprint server that includes a comments function (e.g., bioRxiv) and invite readers to comment. Note that a preprint of this paper has been uploaded to PsyArXiv (https://doi.org/10.31234/osf.io/xaj46), and after it is published, I will continue to update it based on post-publication peer review using the annotation service, Hypothes.is (https://web.hypothes.is/). This process is the exact opposite of the traditional "start with preprint, end with journal" publishing system. In other words, publish in the journal first and then continue to update the post-print. This recommended process may seem odd, but it is a very familiar system that has proven effective in software development. In addition, this post-publication peer review is based on open participation, which means that there is a higher chance of discovering problems than in the usual pre-publication peer review by a few peer reviewers because far more people check the paper. In fact, various frauds, including the STAP cell issue, have been discovered through post-publication peer review, and the ongoing efforts of Elisabeth Bik are well known for the discovery of numerous image frauds.

When you try to implement this in practice, however, you may encounter some practical problems. For example, if you have published many papers, how do you know of the existence of posted comments? This notification issue can be solved in several ways. One would be to set up a system that automatically notifies authors of posts. Readers, too, could be notified when they follow the original paper. This is a common system on social networking sites and very easy to implement. Another suggestion would be to have a dedicated comment submission section in each journal. Creating a new journal dedicated to commentaries is difficult but creating a commentary section in an existing journal is relatively easy.

Furthermore, one may think that compared to software development, the pursuit of truth (the scientific endeavor) is not always as clear-cut (e.g., not using the optimal analysis does not necessarily suggest that the results are false, or hypotheses formation, or induction and interpretation of the results, etc.). Indeed, the scientific effort is not necessary for a definite solution or demand. Therefore, it is certainly unclear how well the post-publication update process could work. Rather, that is why I recognize the great importance of leaving room for papers to be updated after publication. The vaguer the destination, the more we

need to know exactly where we are at this point in time to even decide which direction to go in. For science as well, the literature that comprises it must continue to be updated to its proper state. Moreover, for both software and papers, released material may have initial failures. Therefore, unless the latter has a route with which to deal with failure, it is obvious that knowledge will not be properly accumulated, and science cannot be self-correcting. It is common sense that peer review is not a perfect error detector. Effective error detectors are the large number of people, including early-career researchers, who were not involved in a pre-submission peer review [34]. The error detector alerts will be triggered through post-publication peer review. Reproducibility of science is enhanced when the post-publication peer review is combined with openness.

### 4.2. Open Recommendation

Readers can take even casual ratings, endorsements, and recommendations of a paper as available information about its credibility. The easiest way to accrue this feedback is to post on social networking sites. Although the number of likes on the post may be biased and provide only a rough assessment of its relevance, it is an indicator for evaluating the paper. One academic service that supports this exposure with feedback is Plaudit (https://plaudit.pub/), an open recommendation service through which ORCID-accredited researchers can freely endorse papers. PsyArXiv, a preprint server, was one of the first to partner with this service.

### 4.3. Non-Reviewed Treatment

If a paper published in a predatory journal (or even an invited paper, etc., in a non-predatory journal) is not peer-reviewed, the author can demonstrate their integrity by including this information in the unrefereed paper section of their CV. This strategy has nothing to do with the quality of the paper itself, but it does enhance the author's credibility as a researcher. Although the journal may be predatory, if the paper has been published, it would be inappropriate to include it in the preprint section. As noted above, merely erasing a predatory journal paper from a CV is a type of achievement falsification, and its inclusion in the appropriate section (other than the peer-reviewed papers and preprints sections) of a CV is indeed desirable.

What I have discussed so far are just measures on the individual level. However, what should we do if the (predatory) journal refuses to add bona fide peer reviewers or disclose the review content?[4] At the individual level, we have no further measures that we can take with the journal. I have recommended countering this by inserting links to OSF and PsyArXiv in the text, but not all readers will act in accordance with our intentions to that extent. In that case, it would still be difficult to protect the credibility of an article that has been published in a predatory journal. I would therefore like to expect interventions at the institutional level (the faculties and universities, the publishing companies, and generally the whole publishing process). It would include educational measures such as enhancing digital literacy about publishing. Although not presented here in detail, a survey showed that the victims of predatory journals are widely distributed throughout the world, regardless of economic circumstances [16]. It is clear that the scientific publishing culture itself needs to evolve. If these can create a strong norm of greater flexibility and openness with regard to peer review, then this could mean that concealment of the content of peer-review will be questionable. In other words, even predatory journals will have to be open to peer review, which may one day make it difficult to run a predatory journal itself. For that, further development of community-based public peer-review organizations such as the PCI will also be necessary, as predatory journals may fabricate peer-reviews.

One necessary process for this is the reform of the researcher evaluation. Currently, evaluations from institutions and scientific communities based on the number of publi-

---

4   It is unclear how many journals engage in this kind of act, as each journal has a different strategy, but assuming management with economic interests in mind, it is not surprising that some journals engage in such act.

cations and the impact factor of papers are still in place. However, such rigid evaluation based on a single bibliometric indicator has been criticized (DORA: [35]). In addition, as proposed in the Hong Kong Principles [36], the various good practices of researchers involved in research integrity, such as openness and transparency in all phases of scientific research, should be properly evaluated. This will reduce the incentive for researchers to use predatory journals to publish as many articles as possible [37].

### 5. Conclusions: Open and Credible Science

All research articles, even those published in predatory journals, can enhance their quality and credibility by making the evaluation process as open and multi-layered as possible. At the same time, researchers themselves may be able to stave off the loss of social and academic reputation by admitting that they themselves have published their articles accidentally in predatory journals and by being transparent about the process. However, it is important to note that these strategies are not specific to papers in predatory journals; they should be applied routinely and can be used to "rescue" a paper, even if it is published in a predatory journal[5] The key argument of this article is that open and credible science should always be practiced as a matter of course by all researchers, regardless of whether they submit to a predatory journal. We should abandon the idea of placing value only on the point of publication of a paper, and the entire process of research practice, from pre-registration to post-print updating, should be upgraded to support and improve the way science accumulates knowledge.

**Funding:** This research was supported by JSPS KAKENHI (16H03079, 17H00875, 18K12015, and 20H04581).

**Acknowledgments:** The author would like to thank the Japanese Society for Cognitive Psychology for its support of the Special Interest Group on the Credibility of Psychology, the main body of activity on this paper.

**Conflicts of Interest:** The author declares no conflict of interest. Especially, although the preparation of this manuscript was greatly assisted by Editage's pre-submission peer review, there is no conflict of interest between the authors and Editage, nor is there any intention to promote Editage's pre-submission peer review service.

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
