# Peer review of "How to Protect the Credibility of Articles Published in Predatory Journals"

_publications, doi:10.3390/publications9010004_

Round 1

Reviewer 1 Report

This is a unique article in the literature on predatory publishing. I strongly agree with the author’s basic premise about openness and transparency but some of the reasoning is a little confusing. I also would like the author to separate a) the role of open and other feedback needed to revise, improve, and then submit an article published in a predatory journal to a non-predatory journal versus b) efforts to document the quality of research published in a predatory journal to institutional evaluators or others.

The conclusion is great, and it would be excellent if the author could assert the same in the introductory text. New, dynamic, and varied approaches to scientific publishing benefit science in terms of reproducibility and building open knowledge.

Distinguish between intentional and unintentional predatory authorship. Be more direct that this article discusses steps to take in the event of unintentional predatory authorship other than withdrawal. Ultimately, would not the author need to prepare a new version of the article in the predatory journal to avoid duplication anyway? That is where open peer review becomes very helpful. I doubt that any university or research center would reward a detected publication in a predatory journal if the author could prove its merit with external reviews. That is a very thorny problem. The literature on academic punishment of predatory publishing continues to be deficient.

The quality of peer review in predatory journals is a moving target and it should be acknowledged that there are too many presumptions here. Any peer review that is not open may be subject to low rigor including “legitimate” journals. In the qualitative research on predatory authorship (Cobey et al. and Kurt, Serhat. “Why Do Authors Publish in Predatory Journals?” Learned Publishing 31, no. 2 (April 2018): 141–47) there are clear indications that peer review occurred, but authors did not provide evidence of their reviews.

The tone of the writing is at times too informal, especially in Sections 3-4. Section 3 needs much more expansion around open peer review and describes a complex and varied entity that has no standard definition. I would also be very interested in learning more about the case of the author of this article and his learning process shared indirectly through the article.

Perhaps the author should point towards the Hong Kong Principles regarding openness and transparency in all phases of scientific research, not just pre-registration. I am not remotely a scientist but protocols for research integrity and transparency are integral to the literature on best practices. Articles in biomedical predatory journals found lacking on reporting protocols (David Moher et al., “Stop This Waste of People, Animals and Money,” Nature News 549, no. 7670 (September 7, 2017): 23).

Here are some comments by line number:

8: Acknowledge deliberate predatory publishing. Predatory journals "often" or "may" prey ...

15-17: explicitly connect reproducibility into most of the suggested measures.

26: it might be helpful to define very briefly "publication bias.”

29: rather than write that a journal's "peer-review filter is extremely broad," it would be better to use the phrase "less rigorous" since some predatory journals have broad scopes to maximize income but some prestigious journals also have broad scopes because of their mission, e.g. PLOSOne.

32. Cobey et al (and others) verify that some predatory journals do perform peer review although the depth of the peer review is unclear and presumably shallow. Cobey, Kelly D, Agnes Grudniewicz, Manoj M Lalu, Danielle B Rice, Hana Raffoul, and David Moher. “Knowledge and Motivations of Researchers Publishing in Presumed Predatory Journals: A Survey.” BMJ Open 9, no. 3 (March 2019): e026516. https://doi.org/10.1136/bmjopen-2018-026516.

41-42. Would suggest removing "Note that this paper was prepared in a hurry because there was no publication fee charged for articles submitted by the early deadline for this special."

56-58. I am not sure if arguing that removing predatory publications from a CV is a form of falsification and only authors who unintentionally publish in a predatory journal and regret it would so whether at the behest of their employer or not. This does not really add to the argument.

64-66. Explain much earlier in the paper the limits of the scope of this article. I presumed the author would discuss withdrawal. Also delineate intentional vs. unintentional predatory authorship as early in the article as possible.

72-109. These approaches are all valuable and if non-scientists read the article, they might want a brief introduction to pre-registration. I would also mention that there are journals that will not publish research disseminated in a preprint. Discuss COS’s TOP or the Transparency and Openness Promotion Guidelines explicitly if possible. Many readers will be unfamiliar with these guidelines. 

119. Expand on the importance of open peer review to mitigate predatory publishing. There are at least 10 kinds of open peer review including open commenting post-publication and you need to explain that open peer review can occur before, during, and after the peer review process. Also explain how most peer review is not open, try to supply more recent data, e.g. only X% of all peer review is open. Ross-Hellauer, Tony. “What Is Open Peer Review? A Systematic Review.” F1000Research 6 (August 31, 2017): 588. https://doi.org/10.12688/f1000research.11369.2. Please somehow briefly discuss open data as also integral to improving transparency and reproducibility as well mitigating predatory publishing.

126-134 “Predatory journals are not averse to peer review, but they are afraid of losing their source of income, submitted papers, by issuing rejections. If they know that their prey intends to publish the paper regardless of the peer review results, they will add reviewers on demand.” This sentence/section is very confusing. Do we have any anecdotes of predatory journals rejecting an article and/or adding additional reviewers? Aren’t at least two, if not three reviewers standard?  

139-140. Great but use the phrase “open evaluation” as this form of review is known.

142-143. No one has ever suggested putting an article that was published in a predatory journal in a repository for post-publication review. I think this is interesting. Shouldn’t the author use the feedback to produce a new paper that is not redundant with the one published in the predatory journal?

182-183. This self-promotion is not framed well and should either be deleted or revised to be more neutral.

194-195. Do predatory journals not share back reviewer comments? Do they reject articles? Readers want to know more. The predatory authorship process is shrouded in mystery.

Reviewer 2 Report

In general, the submission presents a highly relevant train of thought and a situation relavant for many researchers from different disciplines on a global level. However, before the piece can be published, some shortcomings will have to be addressed. The following are suggestions for amendment and improvement.

Even though the piece is an opinion piece or special report articles, some phrases are a bit too informal or misleading, f.i. "innocent researcher" - this probably refers to those with less knowledge or awareness about publishing structures. It is something we also find in other parts of the manuscript, f.i. the wording "punishment" of authors is questionable. 

Regarding the topic predatory journals it is unclear to the reader how the presented approach is superior to better education and literacy aiming at identifying and avoiding predatory journals, and how the measures presented by the piece stand in contrast to this practice. Authors point out that the approach can be used also for non-predatory journals, but it is unclear whether and how the presented measures support quality of science. This could be summarized better at the beginning.

"Reputable journals are likely to be selective" - of content or reviewers?

General, unethical or bad practice can go both ways: from journals but also from researchers. The paper makes it look a bit like it is one-sided process. Can one consider both sides of this coin?

"Note that this paper was prepared in a hurry because there was no publication fee charged for articles submitted by the early deadline for this special issue." This argument should be made clearer - what does it show us? We do not know much about the general timeframes of the deadline, which would be needed to follow this argument. Otherwise this could be a good and critical remark. 

"They may not be able to discriminate in their choice of journal" This is unclear and needs rephrasing.

The term "beloved paper" is a bit unclear also. I understand that the opinion should be provocative, but maybe this does not support the arguments.

"note that I consider this to be misconduct, as it is a type of research achievement falsification" - it should be clarified why this is the case.

The advantages of including predatory journalsls in databases like DOAJ, who may also have quality regulations, are not presented clear enough.

Regarding the main question how to ensure the credibility of a paper, the idea that this is independent from the journal type is interesting. However, the reader would need an better overview about the situation and amount of predatory publishing in different disciplines as a starting point. Who is especially at risk and how prevalent is it? 

The advantage of a commercial preer-review needs further explanation and the dangers and limitations of this approach should be pointed out.

How are updates in the form post-publication peer review going to enhance the quality of the paper? This is unclear.

"merely erasing a predatory journal paper from a CV is a type of achievement falsification" - Are there more detailed arguments to outline this claim?

What is the advantage of the author's suggestions compared to, say, enhancing digital literacy regarding the selection of publication venues for papers? And would this not be another important aspect? This needs to be pointed out for conveyance of a convincing argument. 

Round 2

Reviewer 1 Report

The revisions have greatly improved the article and with another round of revisions, I think it should be closer to publication. I have marked up a MSWord copy of the article in the interest of my time and I will only address items for improvement below. Chiefly, I have concerns about insufficient nuance in discussing various topics including peer review and predatory publishing. 

Overall, the introduction needs some strengthening. I felt the discussion of peer review doesn't clearly explain that when a manuscript is rejected, it may be because it is out of scope for the journal. Some predatory journals address this issue by offering very wide scopes but we know that more specialized predatory journals are increasing in numbers. Journals are not so black and white and while predatory journals represent an extreme, there are lower caliber journals that authors may turn to after being rejected. Commercial publishers are known to deliberately move an article from a more prestigious journal to a lesser one under its umbrella in order to keep the content and/or the article processing charge under its purview.  

We also don't know enough about what part of predatory publishing is represented by previously rejected articles although Oermann, M. H., Nicoll, L. H., Chinn, P. L., Ashton, K. S., Conklin, J. L., Edie, A. H., ... & Williams, B. L. (2018). Quality of articles published in predatory nursing journals. Nursing Outlook66(1), 4-10 discusses nursing predatory journal authors as having been previously rejected. Kurt's qualitative research points to fear of rejection as opposed to actual rejection as driving author submission to predatory journals. Cobey (Cobey, Kelly D, Agnes Grudniewicz, Manoj M Lalu, Danielle B Rice, Hana Raffoul, and David Moher. “Knowledge and Motivations of Researchers Publishing in Presumed Predatory Journals: A Survey.” BMJ Open 9, no. 3 (March 2019): e026516. https://doi.org/10.1136/bmjopen-2018-026516.) found the following when they surveyed authors: about 1/3 had been previously rejected and of that group, the most were rejected twice (43%) (p. 7).

The author's arguments about fraud/misconduct occurring when authors who unintentionally publish in a predatory journal removing the publication from their CV are also too polarizing. Many/most authors are unable to withdraw their work once which to me further supports that this argument requires more compassion. I think this decision should be a reflection of a) the author's institutional policies and values and b) the author's own feelings about the quality of the publication. I agree that placing predatory publications, if indeed they received no or little substantive peer review, should be placed as unrefereed if the author desires to list the publication. 

I also would like to see some evidence about predatory journals adding reviewers on demand. This notion is entirely foreign to my knowledge on the topic. 

Round 3

Reviewer 1 Report

Line 12- Still not seeing dicussion of "topping up reviewers" and I continue to be unfamiliar with this terminology

Line 39- "henceforth omitted" - I’d remove this phrase which makes the sentence a little awkward and revise to ... predatory journals and publishers can profit ...

Line 59- Refund?

Lines 61-70- I would suggest reordering some of the text and the footnotes here. Although your final sentence is meaningful and should stay as is, I would remove the sentence that begins with “Hiding achievements” since the second footnote persists in arguing that omission is fraudulent.

I would also not assert in footnote 2 “However, evidence of publication in predatory journals is important for confirming the ethics of the researcher, and this information should not be omitted.” Since there are many innocent authors who are not aware that they published in a possibly predatory journal and for whatever reasons, no concerns were raised at their institution. Ethics implies intent! We cannot presume a) that there was no peer review and b) that the publication itself wasn’t of adequate quality.

You want to argue that more authors who have published in predatory journals should be transparent about what happened but not necessary judge the many authors who may not want to get feedback and post-publication review/resubmit the work/share the work openly. I would be more understanding that different authors will feel ... differently.   

Line 65-66- Consider more formal language: “The author of this article feels that ... “

Line 67- "Scholarly Production": “Scholarly production” (which would include conference presentations) instead? Achievements are presumed to be positive

Line 140-142- Let’s remember that your audience consists of people who accidentally submitted and published with a predatory journal. They were not aware of the journal’s reputation or quality prior to or even during submission.

Line 178-179 (Section 3.2): If true predatory journals exist chiefly to make money at the expense of rigor, they would not want to spend more time and energy on improving any manuscript beyond a bare minimum. If the journal invests considerable time and energy in improving the manuscript, then we can presume the journal may not be truly predatory and is instead lower quality, amateurish or simply seeking to find its place in the publishing ecosystem.

Lines 255-256: "disclose the rveiew content": Be clearer. Do you mean provide feedback to the author or offer an open review? I am not aware that any predatory journal has ever offered an open review. We can presume many predatory publishers push out the article as soon as the APC is received ...